# Cutaneous Metastasis of Endometrial Cancer and Long-Term Survival: A Scoping Review and Our Experience

**DOI:** 10.3390/diagnostics13152603

**Published:** 2023-08-04

**Authors:** Alexandra Nienhaus, Rahavie Rajakulendran, Elena Bernad

**Affiliations:** 1Doctoral School, “Victor Babes” University of Medicine and Pharmacy, EftimieMurgu Square, No. 2, 300041 Timisoara, Romania; alexandra.nienhaus@umft.ro; 2Department of Obstetrics and Gynaecology ‘Augusta Krankenanstalt’ Bochum, Bergstr. 26, 44807 Bochum, Germany; rahavie.r2000@yahoo.de; 3Department of Obstetrics and Gynecology, Faculty of Medicine, “Victor Babes” University of Medicine and Pharmacy, 300041 Timisoara, Romania; 4Clinic of Obstetrics and Gynecology, “Pius Brinzeu” County Clinical Emergency Hospital, 300723 Timisoara, Romania; 5Center for Laparoscopy, Laparoscopic Surgery and In Vitro Fertilization, “Victor Babes” University of Medicine and Pharmacy, 300041 Timisoara, Romania

**Keywords:** endometrial carcinoma, metastasis, long-term survival, soft tissue metastasis, cutaneous metastasis, 5-year overall survival

## Abstract

Background and Objectives: Cutaneous and soft tissue metastases of endometrial cancer are rare. This review aims to examine the prevalence of cutaneous metastasis, the diagnosis and treatment options, and the impact of cutaneous metastasis of endometrial cancer on overall survival. We also present a particular case with a long-term overall survival. Materials and Methods: A systematic literature search was conducted on PubMed and PubMed Central using the following keywords: endometrial carcinoma metastasis, cutaneous metastasis, and five-year overall survival. Results: We identified 326 results and checked their titles for eligibility. There were eight studies included. We also presented a case of a 51-year-old woman with cutaneous metastasis and a large soft tissue metastasis with a prolonged overall survival of about 13 years after the appearance of the first cutaneous metastasis. Conclusions: This paper highlights the importance of skin evaluations in patients diagnosed with endometrial cancer. Healthcare providers must consider the possibility of cutaneous metastasis localization in patients with endometrial cancer to assign the correct stage and apply the appropriate treatment to increase long-term survival.

## 1. Introduction

The most prevalent gynecologic cancer is endometrial cancer [1,2,3]. In 2018, there were 121,600 new cases of this cancer diagnosed in Europe, roughly 6.6% of all malignancies found in women. A total of 29,600 gynecologic-cancer-related deaths are predicted to occur each year, accounting for over 3.5% of all cancer-related deaths [4].

Mostly post-menopausal women are affected by endometrial cancer; it is typically diagnosed in women aged 60 or older [5,6,7]. In women under the age of 45, it is unusual [8,9]. Black women are more prone to develop this cancer than white women, and they also have a higher mortality rate from it [10].

In the US, young women aged 20–29 saw a rise in endometrial cancer from 0.6 per 100,000 in 2001 to 1.2 per 100,000 in 2017 (APC 3.6, 95% CI 2.9–4.4), while women aged 30–39 saw an increase from 4.6 per 100,000 in 2001 to 7.5 per 100,000 in 2017 (APC 3.0, 95% CI 2.7–3.3) [11,12].

Rarely, metastasis develops in the brain, bones, liver, adrenal glands, extra-abdominal lymph nodes, and soft tissue [11,13,14,15]. It is beneficial for imaging specialists and radiologists to be able to identify common and uncommon early atypical metastatic sites.

Skin metastasis from endometrial cancer is rare [16], with a reported frequency of 0.8%. After the discovery of cutaneous metastasis, the mean life expectancy for endometrial cancer patients is reported to be between 4 and 12 months, which is associated with a dismal prognosis. The amount of time between diagnosis and the onset of cutaneous recurrences is a factor that affects survival [17,18,19].

The majority of patients (77%) have an early-stage disease with a decent overall survival (OS) at the time of diagnosis. The 5-year survival rate falls from 95% for localized disease to about 20% after the illness has spread [20]. Patient prognosis is improved by early discovery and treatment.

## 2. Review of Available Evidence on Cutaneous Metastasis of Endometrial Cancer

### 2.1. Review Approach and Methods

In this research study, we wanted to establish a theory regarding the reasons behind long overall survival despite the presence of large cutaneous metastases. Our scoping review aim is to examine the prevalence of cutaneous metastasis, the diagnosis and treatment options, and the impact of cutaneous metastasis of endometrial cancer on overall survival. We also wish to present our experience of a case of the long-term survival of a patient with cutaneous metastasis of endometrial cancer. The research was conducted in accordance with the Joanna Briggs Institute Reviewers’ Manual for Scope Reviews and the PRISMA Extension for Scoping Reviews (PRISMA ScR) criteria. The PICO framework was used to guide the search whenever possible: (P) participants—women diagnosed with endometrial cancer; (I) investigated condition—presence of the cutaneous metastasis; (C) comparison—no cutaneous metastasis; (O) outcome—overall survival.

The protocol for this scoping review was registered on the International Platform of Registered Systematic Review and Meta-analysis Protocols (INPLASY), unique ID: INPLASY202360036.

We performed a systematic search of the current literature on Pubmed and Pubmed Central. For this search, we used the following search keywords: endometrial carcinoma, endometrial adenocarcinoma, metastasis, long-term survival, 5 years survival, soft tissue metastasis, cutaneous metastasis, 5 years overall survival. We combined these keywords with the Boolean operator “AND” as follows: endometrial carcinoma metastasis AND longterm survival, endometrial carcinoma AND metastasis AND 5 years survival, endometrial carcinoma AND soft tissue metastasis, endometrial carcinoma AND cutaneous metastasis, endometrial carcinoma AND 5-year overall survival, endometrial adenocarcinoma AND soft tissue metastasis, endometrial adenocarcinoma AND cutaneous metastasis, endometrial adenocarcinoma AND 5-year overall survival, and endometrial adenocarcinoma AND longterm survival. To narrow down our search, we limited the study selection to studies published between 2003 and 2023. Additionally, we used the following filters: case report, comparative study, meta-analysis, review, systematic review, humans, English, German, 2003–2023.

The studies were examined by two separate researchers (Nienhaus and Rajakulendran), who screened the articles and excluded duplicates in the first stage. Then the abstracts of all potentially relevant papers were individually assessed for their suitability for our review.

We examined the following topics: the time from the initial diagnosis to cutaneous metastasis, the initial operative therapy, the localization of the metastasis, the FIGO stage at the initial diagnosis of the endometrial carcinoma, the therapy of the initial diagnosis, the therapy of the metastatic disease, and the time from diagnosis of the cutaneous metastasis to death. For this, we created different tables to represent our results more clearly.

After conducting this survey, we identified 326 results and checked the titles for their eligibility. Our inclusion criteria were case reports, reviews, endometrial carcinoma, endometrial adenocarcinoma, and cutaneous metastasis. We excluded 297 studies dealing with other topics such as the relevance of lymphadenectomy, other forms of endometrial cancer, other locations of metastasis, other carcinomas, the recurrence of endometrial cancer, hormone therapy, radiotherapy, chemotherapy, different treatments of the endometrial carcinoma, anemia, immunological therapy, prognostic factors, imaging, other diseases, brachytherapy, influence of the physician, endometriosis, racial disparities, and prediction of recurrence. Thus, we determined that 29 of the 326 studies were suitable for this review. In the next step, we moved duplicate variants and obtained a list of 16 studies. The last screening was an examination of the full texts. Finally, we included 5 studies and excluded 11 studies. In addition, we viewed the primary content of the 16 studies we included before and selected three of them as suitable. Therefore, there were a total of 8 studies in our review (Figure 1).

### 2.2. Data Synthesis

Our goal in performing this study was to develop a theory underlying the prolonged overall survival in spite of significant cutaneous metastases. Examining the frequency of cutaneous metastasis, the available diagnoses and therapies, and survival times is the main objective of this scoping review. 

## 3. Results

We found eight case report studies of cutaneous metastasis of endometrial cancer. The cohorts included patients between 52 and 73 years old. The histological type of the metastasis was endometroid adenocarcinoma in six cases, carcinosarcoma in one case, and leiomyosarcoma in another case. The time from the diagnosis of endometrial cancer to the appearance of cutaneous metastasis was between 2 months and 3 years (Table 1).

The location of cutaneous metastasis varied. In two studies, a metastasis was located on the skull and two studies showed metastasis on the vulva. Metastases were also located on the navel, trunk, breasts, flank, and lower leg. The initial operation performed for endometrial cancer depended on the FIGO stage, with total abdominal hysterectomy and bilateral salpingo-oophorectomy with or without a pelvic or para-aortic lymphadenectomy. Some patients only underwent lymph node sampling. When a lymphadenectomy was performed, in most cases, metastasis in the lymph nodes was observed (Table 1).

In four of the studies, the initial FIGO stage was I; in three others, the FIGO state was III; and in one case, the FIGO stage was IVB with spread of the disease to the pelvic lymph nodes (Table 2).

We can observe that after the appearance of cutaneous metastasis, the overall survival rate is poor, at between 2 weeks and 14 months. In our case, there was a long overall survival of over 13 years. The therapy had no significant effect on the overall survival rate. Chemotherapy was performed in five studies and radiotherapy in two cases. Hormone therapy was performed in three cases, and, in one case, no therapy was performed due to the poor general condition of the patient. The appearance of cutaneous metastasis was never the only sign of metastasis. In most cases, there were also signs of pulmonary, intraabdominal, and lymph node metastasis and pleural effusion (Table 2).

## 4. Case Presentation

We present the case of a 51-year-old woman presented to a hospital in Bochum, Germany, with a diagnosis of endometrial adenocarcinoma pT1b pN1(4/24) M1 (cutaneous, umbilical metastasis) G1-2 L1 R0 in 2008. A hysterectomy/abdominal adnexectomy with excisions of the navel, pelvic, and para-aortic lymphadenectomies was performed. She received after loading radiation three times.

The patient had no medical history except for arterial hypertension. In the family, there was an aunt who received a breast cancer diagnosis at the age of 56 and also a cousin who received a breast cancer diagnosis at the age of 50. Both are maternal relatives.

She returned to Bochum in June 2018 with the following symptoms: on the right ventral chest wall, she had a large tumor with a perforation in the pericardium of approximately 15 × 20 cm, an abdominal inguinal tumor on the right, and an ulcerating tumor on the lower left side of the abdomen. A computed tomography of the thorax and abdomen was performed in 09/2018 (Figure 2 and Figure 3) and a magnetic resonance imaging of the thorax was performed in 06/2018 (Figure 4 and Figure 5).

The histopathological findings (Figure 6) of a probe of the large chest and inguinal tumors show an endometroid adenocarcinoma, G2, with expression of cytokeratins MNF116 and CK7. The reaction to CK20, CDX2, Gata3, and TTFI was negative. The PAX8 marker was positive. The receptor status was positive to estrogen 90% and progesterone 40%. PDL1 was 1%. The intrinsic subtype was a microsatellite stabile tumor.

Chemotherapy with carboplatin/paclitaxel cycle I was initiated on 19 June 2018 and radiotherapy of the abdominal tumor mass, the inguinal tumor on the right side, and the ulcerating tumor on the lower left side of the abdomen was performed between 6 June 2018 and 8 August 2018. The dose of radiation was 50.4 Gy/1.4 Gy. On 15 August 2018, carboplatin/paclitaxelis was administered weekly and the patient had a reaction to the contrast agent under RT RCXT. In 2018/09, we tried to determine the cause of the prolonged pancytopenia that led to the limitation of therapy and thrombocytopenia. The most likely origin of toxicity seemed to be related to the drugs administered after cytostatic therapy. A differential diagnosis of idiopathic thrombocytopenia was not the most likely. A bone marrow diagnosis does not currently indicate myelodysplasia. Between 2018/11 and 05/2019, we switched to megestrol acetate therapy. The treatment was interrupted because the medicine was not available. In 2019/05, thrombosis of the left common iliac vein occurred. Apixaban therapy was initiated in 2019/07. An abdominal CT scan was performed, and modest progress was noted. Treatment with tamoxifen 20 mg/day was initiated. An abdominal ultrasound was performed in 2019/09 and stable disease was confirmed; thus, tamoxifen treatment was continued. On 10 June 2020 MB, progression of the chest disease with a pronounced volume and appearance of skin metastasis were noted. A comprehensive molecular diagnosis showed possible skin metastasis. Therapy with pembrolizumab (no carboplatin and pemetrexed due to limited hematological competence) was initiated. In 2020/09, CT was performed; the results showed new progression and progressive intramammary manifestation of parts of the tumor, which was now spreading to the skin of the right breast. Pembrolizumab therapy was then initiated on 23 September 2020. Several symptoms were suspected as hypertension. In 2019/04, thrombosis of the left common iliac vein was noted; apixaban therapy was initiated. There was a discrepancy between the manifestation of the tumor and the excellent clinical status to date (extremely slow-growing tumor).

The patient developed axillary skin metastasis of the right axilla in 2021 (Figure 7). Intensity-modulated radiotherapy(IMRT/RapidARc) as a palliative therapy with 30/2 Gy and simultaneous boost with 7.50 Gray was performed. The cumulated dose was 37.5 Gray, applied to the right axilla. The patient had no adverse reactions to radiotherapy. Computed tomography of the thorax and abdomen in 01/2021 (Figure 8 and Figure 9) showed the slow growth of the masses compared to 2020.

The patient passed away in 2021.

## 5. Discussion

Although early-stage disease limited to the uterus is found in the majority of women with EC, a sizable portion of cases also have metastatic disease [27].

Our patient was diagnosed with an advanced stage of disease in the lymphatic nodes and had a relapse after 10 years of progression-free survival. This is an unusual finding with large metastasis of the soft tissue from the chest wall including bone destruction and cutaneous metastasis.

### 5.1. Incidence and Risk Factors

Cutaneous metastasis is often a late manifestation of disease. It can also occur as the first manifestation of internal malignancies.

The disease usually spreads locally or by lymphatic dissemination [28,29]. Distant metastases are less common and usually involve the lungs and liver as per hematogenous spread. Marianiet et al. identified a 7.5% incidence of lung and 1.9% incidence of liver metastases in a cohort of 612 endometrial cancer patients. Bone metastasis at the time of diagnosis is extremely rare. When bone metastases are observed, it is usually because the illness has relapsed while being monitored following the initial course of treatment. Interestingly, research has found that, when autopsy results were taken into account, bone metastases were observed in up to 25% of all endometrial carcinomas. The most frequent metastatic sites were the vertebrae [30].

When skin metastases are discovered, they typically signify an extensive spread of the underlying cancer. However, in this instance, endometrial cancer was discovered when it was still treatable with surgery alone. The discovery of cutaneous metastasis concurrent with lung metastasis suggests extensive hematogenous spreading [19].

### 5.2. Risk Factors for Endometrial Cancer

Exposure to endogenous and exogenous estrogen is linked to diabetes, obesity, an early menarche age, nulliparity, a late onset menopause, older age (55 years), and tamoxifen use. It is debatable whether diabetes and endometrial cancer are related [31,32]. Only one of the four cohort studies where adjustments for body mass index (BMI) were undertaken found an independent link between endometrial cancer and diabetes [33].

A history of pelvic radiation is also a risk factor for endometrial cancer [21,34].

The stage of the disease and the involvement of lymph nodes are very strong risk factors. The time for metastasis of the skin ranges from months to a few years.

### 5.3. Clinical Presentation

Most patients are referred to a hospital when skin metastases are observed. In some cases, they were bleeding skin metastasis, but in most cases, they were nodules or skin papules without bleeding or symptoms. Therefore, it is very important to examine the skin of a patient after a diagnosis of endometrial cancer. A histological biopsy should always be performed.

Breast (69%), colon (9%), melanoma (5%), and ovarian cancer (4%) have been shown as the primary tumors that commonly cause metastasis of the skin [19].

Clinically, cutaneous metastasis appears as nodules, ulcers, lesions that resemble cellulitis, bullae, or fibrotic processes [23,35,36].

### 5.4. Image Examination

Due to the appearance of cutaneous metastasis, a reevaluation of the status of health should be performed, as we observed that cutaneous metastasis was not a unique metastasis event. By conducting computed tomography (CT), magnetic resonance imaging (MRI), or positron emission tomography (PET CT), we detected other metastases. and this potentially changes the therapy sequence.

### 5.5. Histological Examination of the Metastasis

In recent years, there have been a lot of new options for diagnosing endometrial cancer and immunohistological findings, presenting new therapy options.

Genetic mutations cause endometrial cancer in about 5% of patients, occurring 10 to 20 years before sporadic cancer, but most endometrial cancer cases (95%) are caused by sporadic (somatic) mutations. Since there is an increasing overlap in the histopathological features of these tumors, molecular analyses (e.g., identification of characteristic translocations and/or mutations) and subtype classification are useful in choosing appropriate therapies [37].

Therefore, the MMI (mismatch repair proteins) and MSI (microsatellite instability) of the tumor should be examined, because immune therapy could be a good option for treatment. A genetic prevalence of endometrium cancer could also be identified [38,39,40,41].

Patients with a significant family history of endometrial and/or colorectal cancer should be referred to genetic counseling and evaluation for “Lynch Syndrome [Hereditary Non-Polyposis Colorectal Cancer]” [42,43,44,45]. This recommendation applies to patients with MMR defects, MSI-stable individuals, patients without screening, and patients without MSI defects. For patients under the age of 50, screening for genetic abnormalities should be taken into consideration [37].

### 5.6. Therapy

Nowadays, there are several therapy options for metastatic endometrial cancer, starting with chemotherapy, hormonal therapy, and immune therapy. For local spread of metastasis, radiotherapy is an option.

Chemotherapy is one of the initial therapies for metastatic disease. In our case, we performed combination therapy with carboplatin/paclitaxel.

For advanced or recurrent endometrial cancer, the combination of carboplatin and paclitaxel is being used more frequently; the response rate is between 40% and 62% and the OS ranges between 13 and 29 months. Carboplatin and paclitaxel were compared to cisplatin, doxorubicin, paclitaxel, and filgrastim (granulocyte colony-stimulating factor) in a phase III trial (GOG 209). According to the trial data, the oncologic results were comparable, but carboplatin/paclitaxel had a better safety and tolerability profile. As a result, many patients now favor the carboplatin/paclitaxel regimen. Docetaxel and carboplatin may be considered for people for whom paclitaxel is counter-indicated [37].

These patients may receive palliative or curative care, depending on previous treatments and the type of recurrence. The rarity of this particular metastatic site has prevented the development of proven effective treatments to date. Treatment plans, however, have been developed based on the knowledge of other recurrent endometrial cancer presentations. Current recommendations call for local excision with one skin metastasis, if practical. The majority of patients receive palliative care, but while chemotherapy and radiotherapy are frequently administered to these patients, they are often ineffective [19].

Hormonal therapy has not yet been proven to be particularly effective in treating endometrial cancer. There is inadequate proof that hormone therapy, regardless of its dosage, type, or inclusion in a combination therapy regimen, increases patients’ chances of surviving advanced or recurring endometrial cancer. The choice to employ any kind of hormonal therapy should be made individually, with the goal of palliating the disease, given the lack of a proven survival advantage and the diversity of patient demographics [30].

In our case, we used megestrol acetate, which was well tolerated. Unfortunately, we had to change the treatment due to the lack of availability of the medicine.

Hormone therapy is mostly employed for advanced or recurrent endometrial cancer [46,47,48,49,50]. If the endometrial cancer is hormone receptor positive or exhibits no clinical signs, it will respond more favorably to this therapy [51].

In patients with endometrioid histologies alone, the role of hormone therapy in recurring or metastatic cancer has been examined. For lower-grade endometrioid histologies, hormonal treatment is often employed, ideally in individuals with a modest tumor volume or a slow growth rate. Megestrol acetate with alternating tamoxifen, medroxyprogesterone acetate/tamoxifen (alternating), everolimus/letrozole combination, progestational agents (such as medroxyprogesterone acetate and megestrol acetate), aromatase inhibitors, tamoxifen alone, or fulvestran are some examples of hormones used to treat recurrent/metastatic disease [37].

## 6. Conclusions

Endometrial primary malignancies metastasize most commonly in the vagina or perineum. Rarely, these tumors metastasize in soft tissues or cutaneously. In most cases, the overall survival rate is poor, from months to only a few years.

Skin metastasis from endometrial cancer is extremely uncommon, despite the fact that endometrial cancer is one of the most common cancers in women. Subcutaneous nodules are a sign of expansive spreading and impending death, indicating that these patients have a terrible prognosis.

In our case, we observed a rare case of a patient with a long survival, with unusual, poor symptoms of a large, severe metastasis of the chest wall with invasion of the thorax. At the first diagnosis of endometrial cancer, the patient presented a metastasis of the navel and metastatic lymph nodes. This is the only case found in the literature review spanning more than 10 years of survival after presenting with cutaneous metastasis.

Examination of the skin in patients with a diagnosis of endometrial cancer even after receiving local or systemic treatment is crucial. Nowadays, we have new therapy options for local or metastatic endometrial cancer in the form of immune therapy. Thus, a histological probe of metastasis is needed when it is diagnosed.

## Figures and Tables

**Figure 1 diagnostics-13-02603-f001:**
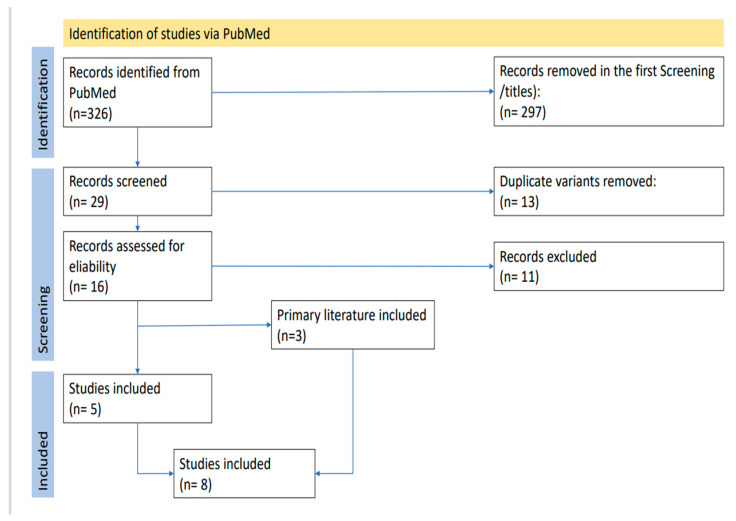
Prisma flow chart with identification of the studies.

**Figure 2 diagnostics-13-02603-f002:**
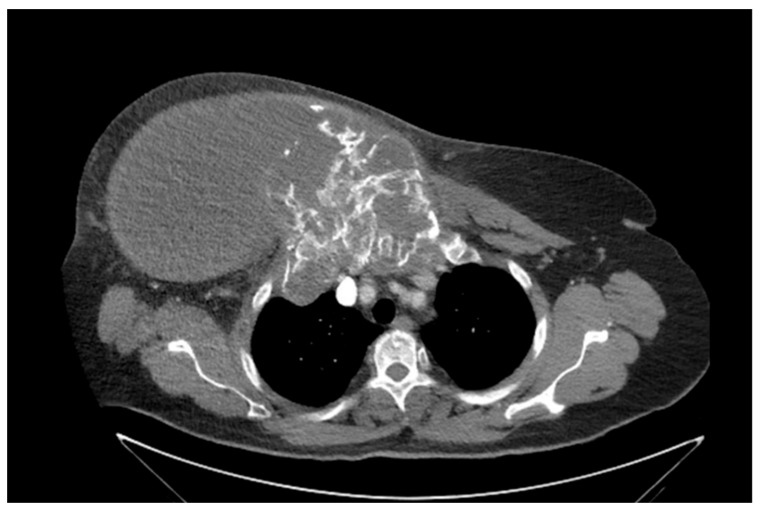
Computed tomography of thorax and abdomen (CT Thx/Abd) in 2018. In this study, we detected a large tumor on the right anterior thorax with calcifications and invasion in the mediastinum.

**Figure 3 diagnostics-13-02603-f003:**
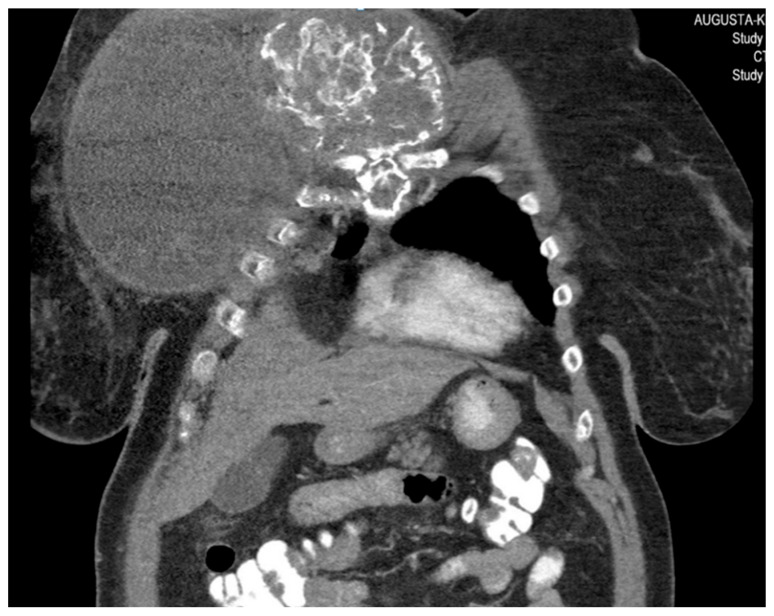
Computed tomography of the thorax and abdomen (CT Thx/Abd) in 2018. There is a large, partially calcified soft tissue structure of the right breast with central hypodensity and possibly mucinous parts. The image shows infiltration of the thoracic wall and transfer to the mediastinum, with destruction of the corpus and manubrium sterni and infiltration of the musculus pecotralis major and the musculus serratus anterior on the right. Additionally, destruction of the adjacent medial clavicle on the right and of the first rib medial on the right was observed. The formation directly infiltrates the mediastinum, with infiltration of the pericardium and direct contact with the aortic arch. The maximum extent of this conglomerate was 25.2 × 14.8 × 10.0 cm (VU 22.7 × 16.6 cm axial; not fully recorded cranially). There was no flow obstruction of the right-side vessels of the upper thoracic aperture.

**Figure 4 diagnostics-13-02603-f004:**
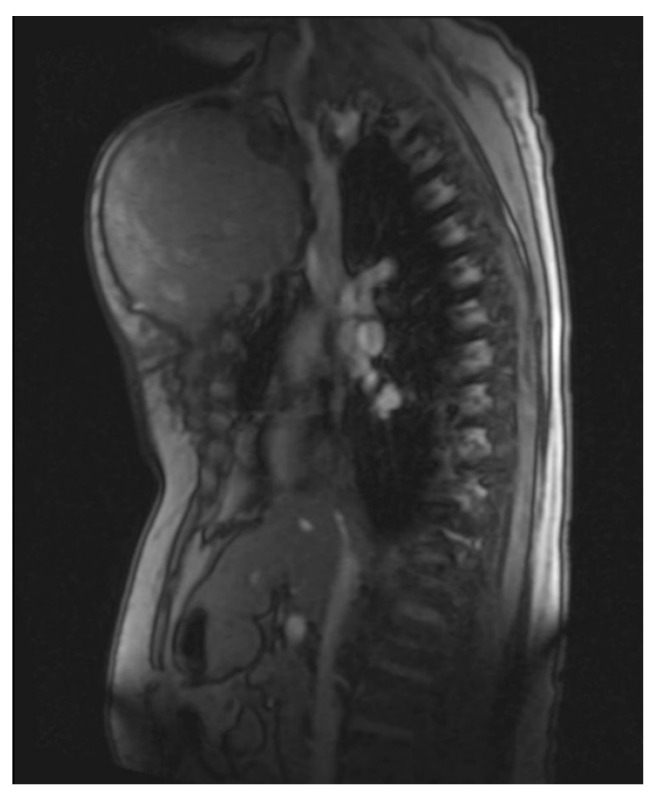
Magnetic resonance imaging (MRI 2018) showing the large tumorous masses on the anterior thoracic wall, infiltration of the mediastinum per continuitatem, and contact with the aorta.

**Figure 5 diagnostics-13-02603-f005:**
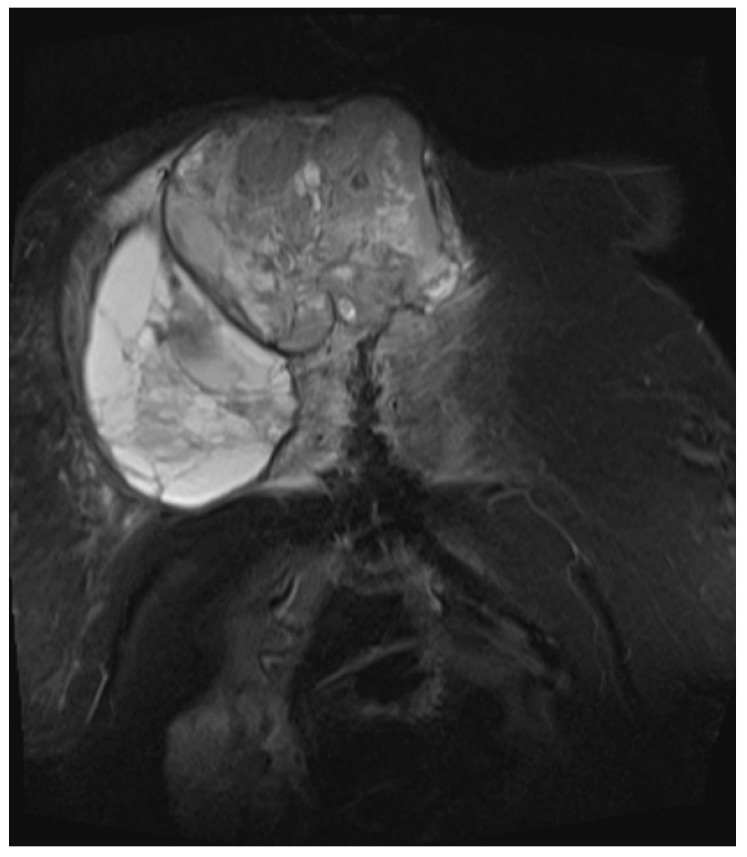
Magnetic resonance imaging (MRI 2018) showing the bis soft tissue structure, partially calcified and infiltrating the sternum.

**Figure 6 diagnostics-13-02603-f006:**
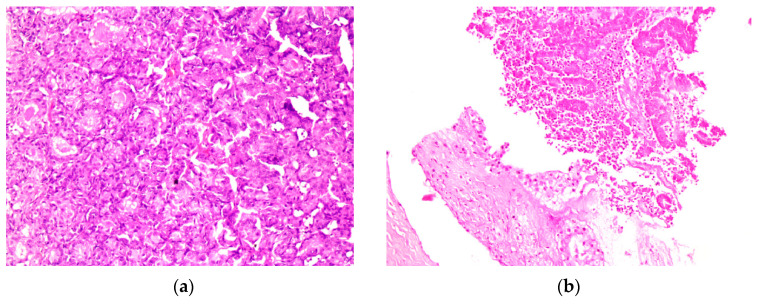
(**a**) HE; 250× magnification under a microscope, metastasis of the rectus abdominis muscle; (**b**) HE 250× magnification under a microscope, invasion of the soft tissue from the endometrial cancer metastasis on the right thorax wall. Abbreviations: HE, hematoxylin -eosin.

**Figure 7 diagnostics-13-02603-f007:**
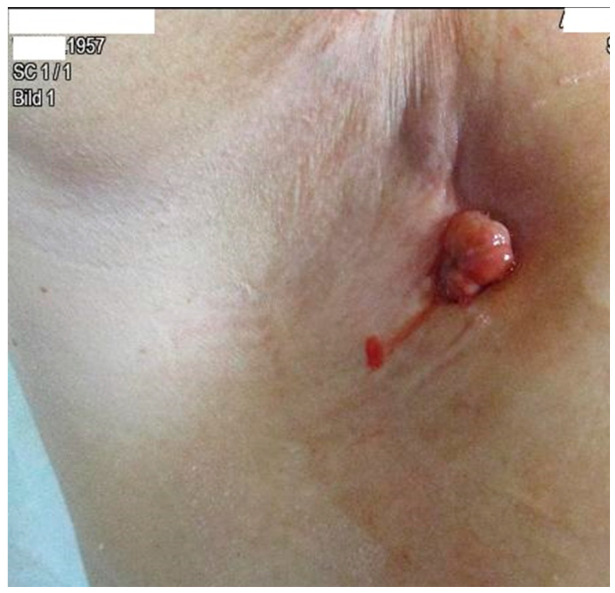
Image of axillary cutaneous metastasis in the right axilla in January 2021.

**Figure 8 diagnostics-13-02603-f008:**
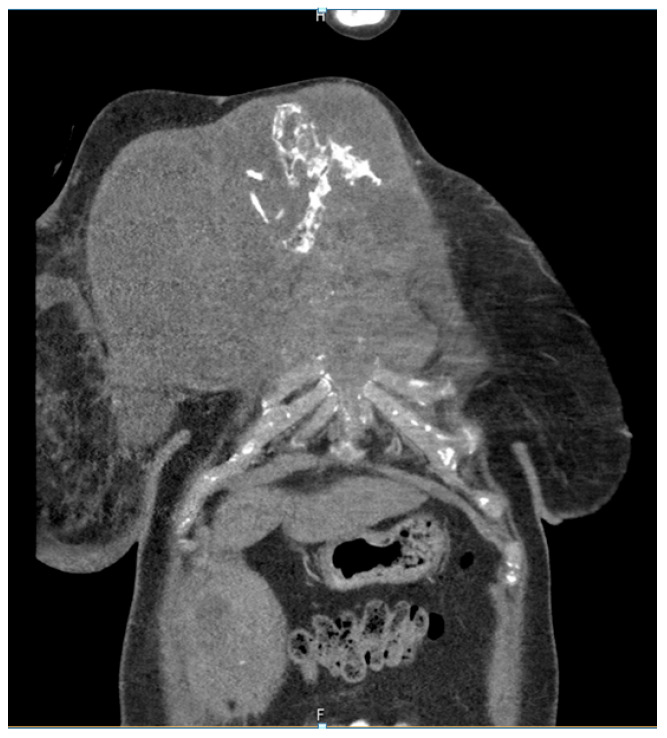
Computed tomography of the thorax and abdomen (CT Thx/Abd) in 2021showing large tumorous masses on the right anterior thoracic wall and right mammary gland with extensive calcification. This extended transversally over approximately 13.4 × 24.9 cm compared to approximately 12.7 × 25.4 cm in the CT scan on 04/20. The tumor formations extend craniocaudally over approximately 17.6 cm, compared to up to approximately 16.1 cm in 04/20 and up to approximately 17.0 cm in 09/20. Includes osteosynthesis material at the HWK 5/6 level.

**Figure 9 diagnostics-13-02603-f009:**
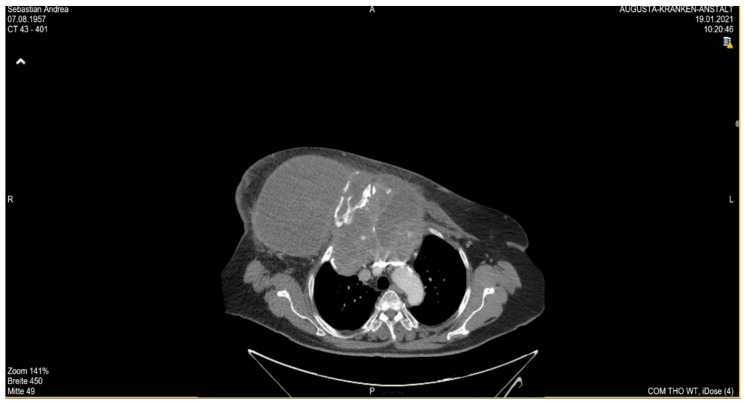
Computed tomography of the thorax and abdomen (CT Thx Abd) in 2021 showing the continued spread in the cervicothoracic junction. Osteolytic destruction of the medial right clavicle, the first to third right ventral ribs, the manubrium sterni and corpus sterni, as well as the first and second ribs on the left was observed. Infiltration of the caudal costal cartilage transitions was also observed.

**Table 1 diagnostics-13-02603-t001:** Summary of authors, year of publishing, study type, age of included patients, histological findings of primary disease, time from first diagnosis to the appearance of cutaneous metastasis, operation performed upon primary diagnosis of endometrial cancer, and the location of cutaneous metastasis.

Authors	Year	Study Type	Age	Histological Findings	Time from First Diagnosis to Cutaneous Metastasis	Initial Operative Therapy	Location of Metastasis
Atallah et al. [17]	2014	CR	62	E.A.	3 Years	Total hysterectomy, bilateral salpingho-oopherectomy, and pelvic lymphadenectomy	Located at the site of thelaparotomy extending down to the vulva
Clairwood, MD [21]	2016	CR	57	Carcinosarcoma	6 Months	Total hysterectomy, bilateral salpingho-oopherectomy, and pelvic and paraaortic lymphadenectomy	Asymptomatic noduleslocated on her left lower cheek, back, and flank
Giovanna Giordano [22]	2005	CR	66	E.A.	8 Months	Total hysterectomy, bilateral salpingo-oophorectomy, and selective pelvic lymph node sampling	Vulvar posterior commissure
Nikolaos Barbetakis [23]	2009	CR	52	Leiomyosarcoma	2 years	Total hysterectomy, bilateral salpingo-oophorectomy, and pelvic lymphadenectomy	Cutaneous skull metastasis
Baydar [24]	2005	CR	58	E.A.	19 Months	No information specified	Cutaneous metastasis ofinitial operation site
Fatima Zahra El M’rabet [19]	2012	CR	72	E.A.	6 Months	Total abdominal hysterectomy, bilateral salpingo-oophorectomy, and pelvic and paraaortic lymph node sampling	Subcutaneous nodules located on her trunk, extremities, and scalp
Yuan Fan [25]	2021	CR	54	E.A.	22 Months	No information specified	Cutaneous breast metastasis
Stonard MC [26]	2003	CR	73	E.A.	2 Months	Total abdominal hysterectomy, bilateral salpingo-oophorectomy	Skin lump on her leftlower leg
Nienhaus	2023	CR	51	E.A.	At primary diagnosis	Hysterectomy/abdominal adnexectomy with excision of the navel, pelvic and paraaortic lymphonodectomy	Cutaneous, umbilical metastasis

Abbreviations: CR, case report; E.A., endometrioid adenocarcinoma.

**Table 2 diagnostics-13-02603-t002:** Summary of study: authors, year of publication, study type, age of included patients, FIGO stage at primary diagnosis, therapy performed after surgery at primary diagnosis, treatment of metastatic disease, and time until death after diagnosis of cutaneous metastasis.

Authors	Year	Study Type	FIGO Stage at First Diagnosis	Therapy after Initial Diagnosis	Treatment of Metastatic Disease	Time until Death after Diagnosis of Cutaneous Metastasis.
Atallah et al. [17]	2014	CR	I B	Extern pelvis radiation	Four cycles of chemotherapy with etoposide and cisplatin	5 months after chemotherapy
Clairwood, MD [21]	2016	CR	III C1	Carboplatin/taxol and whole pelvis radiation	Radiotherapy	3 months
Giovanna Giordano [22]	2005	CR	III C	None, due to fragility	Hormone therapy	14 months
Nikolaos Barbetakis [23]	2009	CR	FIGO III	CTX, RTX	Operation, chemotherapy	Still alive after 8 months of therapy (at the time of article presentation)
Baydar [24]	2005	CR	-	RTX, CTX	Combination chemotherapy	After two cycles of chemotherapy
Fatima Zahra El M’rabet [19]	2012	CR	FIGO IC	N	None due to poor general condition	2 weeks
Yuan Fan [25]	2021	CR	FIGO IB	None	Chemotherapy, immunotherapy, and hormone therapy	7 months
Stonard MC [26]	2003	CR	FIGO IC	None	No information	No information
Nienhaus	2023	CR	FIGO IVB	RTX	Chemotherapy, immunotherapy, and hormone therapy	13 years

Abbreviations: CR, case report; CTX, chemotherapy; RTX, radiotherapy.

## Data Availability

All the data used in this study are published in the literature.

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
