# Peer review of "Cutaneous Metastasis of Endometrial Cancer and Long-Term Survival: A Scoping Review and Our Experience"

_diagnostics, 2023, doi:10.3390/diagnostics13152603_

Round 1
Reviewer 1 Report
Nienhaus A et al. reviewed the prevalence, diagnosis and long-term survival of cutaneous metastasis of soft tissue endometrial cancer. Authors have provided an insightful analysis of clinical cases of endometrial cancer and its cutaneous metastasis. But, they could write well the section 4, where they presented a case report. While reading the ‘case report,’ the authors seem to be presenting their work at a seminar. Authors must prepare the manuscript to fix the grammatical mistakes. For example, ‘The treatment is interrupted because the medicine is not available.’
1. Rewrite the sentence in line- 53.
2. Authors have presented figures sequene inappropriately, figure 6-7 is presented before figure 2.
3. Correct the spelling of ‘treatament’ in line 335.
Nienhaus A et al. reviewed the prevalence, diagnosis and long-term survival of cutaneous metastasis of soft tissue endometrial cancer. Authors have provided an insightful analysis of clinical cases of endometrial cancer and its cutaneous metastasis. But, they could write well the section 4, where they presented a case report. While reading the ‘case report,’ the authors seem to be presenting their work at a seminar. Authors must prepare the manuscript to fix the grammatical mistakes. For example, ‘The treatment is interrupted because the medicine is not available.’
1. Rewrite the sentence in line- 53.
2. Authors have presented figures sequence inappropriately, figure 6-7 is presented before figure 2.
3. Correct the spelling of ‘treatament’ in line 335.
Author Response
1. Rewrite the sentence in line- 53.
Answer: The sentence was rewrited. Thank you for this suggestion.
2. Authors have presented figures sequence inappropriately, figure 6-7 is presented before figure 2.
Answer: We rearranged the figures numbering in the review
3. Correct the spelling of ‘treatament’ in line 335.
Answer: the spelling was corrected. Thank you for the comment
The article was sent for check with the MDPI English Editor.
Reviewer 2 Report
Cutaneous Metastasis in Endometrial Cancer and Long Term Survival: A Scoping Review
Endometrial cancer is one the most prevalent gynecologic cancer mostly affecting post-menopausal women. The majority of patients have an early-stage disease, the overall survival is drop significantly if the disease has spread. Early detection and treatment is imperative. Metastasis, which are rarely, develop in the brain, bones, liver, adrenal glands, extra-abdominal lymph nodes, and soft tissues. Skin metastasis from endometrial cancer are uncommon. The authors performed a review to examine the prevalence of cutaneous metastasis, diagnosis and treatment options, and impact of cutaneous metastasis in endometrial cancer on overall survival. This is clinical relevant and could guide treatment in practice and further research on this topic.
Title: The title is well chosen, resembling the performed study.
Overall: The paper is well written and attractive to read from a reader point of view. The aim of the study is NOT well emphasized and explained in the manuscript. First a systematic review is presented and all of a sudden a case is reported.
Abstract: Should be adjusted on basis on the provided comments and suggestions
Introduction :
The introduction section is attractive to read and emphasizes the reason for conducting the study.
No comments on this section.
Review of available evidence on cutaneous metastasis in endometrial cancer
1. Although the section is well written and attractive to read, the authors report that a systematic review of the literature was conducted. To a surprise, a case report is also reported. This was a surprise for me as a reader. This should be reported earlier in this section that besides a systematic review, also a case report is presented.
2. In this section result are presented, number of studies and the recruitment result. These result should be reported in the result section. This helps the reader in understanding and interpretation of the presented results.
Results
This section is well written and very pleasant to read. Despite, three are some points which could be improved.
3. Table 1 and 2 should be merged as the majority of columns are identical in the tables. The use of abbreviations, e.a. for received treatment, might help.
4. Table 3 and 4 could also be merged to one table, making it easier for the reader.
Case report
The case report is well presented, with thorough description of the case, which is excellent.
5. The presentation of a case report was unexpected, as this was reported in the title, in the introduction or earlier in the article. So , this case report is there out of a sudden. The authors should report the presentation of a case , preferably in the title and much earlier in the article.
6. The current case should have been included in the recruited case from the performed systematic review, as now this knowledge is not incorporated in the earlier results.
Discussion
The discussion section is well written and easy to read.
7. As a reader , I wonder if the presented statements and conclusions are based on the systematic review and if the presented case has been included in the presented conclusions?
Author Response
1. Although the section is well written and attractive to read, the authors report that a systematic review of the literature was conducted. To a surprise, a case report is also reported. This was a surprise for me as a reader. This should be reported earlier in this section that besides a systematic review, also a case report is presented.
Answer: we wrote now in the text that we show also a case presenting our experience to make it better undestandable.
2. In this section result are presented, number of studies and the recruitment result. These result should be reported in the result section. This helps the reader in understanding and interpretation of the presented results.
Results
This section is well written and very pleasant to read. Despite, three are some points which could be improved.
3. Table 1 and 2 should be merged as the majority of columns are identical in the tables. The use of abbreviations, e.a. for received treatment, might help.
Answer:We merged now the table 1 and 2. Thank you for the advise.
4. Table 3 and 4 could also be merged to one table, making it easier for the reader.
Answer:We merged now the table 3 and 4. Thank you for the advise.
Case report
The case report is well presented, with thorough description of the case, which is excellent.
5. The presentation of a case report was unexpected, as this was reported in the title, in the introduction or earlier in the article. So , this case report is there out of a sudden. The authors should report the presentation of a case , preferably in the title and much earlier in the article.
Answer: we reported now this case report as our experience in the title. Thank you for this good comment.
6. The current case should have been included in the recruited case from the performed systematic review, as now this knowledge is not incorporated in the earlier results.
Answer: the knowledge was incorporated in the results now.
Discussion
The discussion section is well written and easy to read.
7. As a reader , I wonder if the presented statements and conclusions are based on the systematic review and if the presented case has been included in the presented conclusions?
Answer: the presented case was included in the conclusions now.
Round 2
Reviewer 1 Report
Authors have addressed my concerns.
Author Response
Thank you for the time allocated and the effort you made to review the article.Reviewer 2 Report
The authors have revised the article and thereby improved the quality of the article.
May I compliment the authors with this achievement.
Author Response
Thank you for the time allocated and the effort you made to review the article.